# Design and Validation of qPCR-Specific Primers for Quantification of the Marketed *Terfezia claveryi* and *Terfezia crassiverrucosa* in Soil

**DOI:** 10.3390/jof8101095

**Published:** 2022-10-17

**Authors:** Francisco Arenas, Asunción Morte, Alfonso Navarro-Ródenas

**Affiliations:** 1Departamento de Biología Vegetal (Botánica), Facultad de Biología, Universidad de Murcia, CEIR Campus Mare Nostrum (CMN), Campus de Espinardo, 30100 Murcia, Spain; 2Forest Science and Technology Centre of Catalonia (CTFC), Carretera de Sant Llorenç de Morunys, Km 2, 25280 Solsona, Spain

**Keywords:** ITS, *Terfezia claveryi*, *Terfezia crassiverrucosa*, qPCR, desert truffles, rDNA, ascomycetes, mycelium

## Abstract

Desert truffle crop is a pioneer in southeastern Spain, a region where native edible hypogeous fungi are adapted to the semiarid areas with low annual rainfall. *Terfezia claveryi* Chatin was the first species of desert truffle to be cultivated, and has been increasing in recent years as an alternative rainfed crop in the Iberian Peninsula. However, its behaviour in the field has yet not been investigated. For this purpose, specific primers were designed for the soil DNA quantification of both *T. claveryi* and *Terfezia crassiverrucosa* and a real-time qPCR protocol was developed, using the ITS rDNA region as a target. Moreover, a young desert truffle orchard was sampled for environmental validation. The results showed the highest efficiency for the TerclaF3/TerclaR1 primers pair, 89%, and the minimal fungal biomass that could be reliable detected was set at 4.23 µg mycelium/g soil. The spatial distribution of fungal biomass was heterogeneous, and there was not a direct relationship between the quantity of winter soil mycelium and the location/productivity of desert truffles. This protocol could be applied to tracking these species in soil and understand their mycelial dynamics in plantations and wild areas.

## 1. Introduction

*Terfezia claveryi* Chatin is an edible mycorrhizal hypogeous fungi belonging to the *Pezizaceae* family that establish mycorrhizal symbiosis with some plants of the *Helianthemum* genus [1]. Its natural habitats are arid and semiarid environments with a low annual rainfall inputs, mild winters, and warm summers, mainly encompassing countries of the Mediterranean geographical region [2,3]. *T. claveryi* was the first desert truffle to be cultivated [4], and it is known to be one of the most appreciated desert truffle species on the market [5], together with other known desert truffles (mainly *Terfezia boudieri* Chatin, *Tirmania nivea* (Desf.) Trappe and *Tirmania pinoyi* (Maire) Malençon [6]). In addition, desert truffles are not only an important economic resource, but contain high nutritional and antioxidant properties [7,8], including bioactive compounds with potential health benefits such as antimicrobial, anti-inflammatory, hepatoprotective, and antitumor activities [9,10,11,12].

Recently, the area cultivated with the desert truffle *T. claveryi* has been increased in semiarid areas of Spain [5,13], becoming an alternative agricultural crop thanks to low water requirements for cultivation [14]. Until now, some abiotic factors or agroclimatic parameters associated with plant management and the control of fungal fruiting have been studied [14,15,16]. Although this knowledge on mycorrhizal plant phenology could helped to stabilise annual fluctuations in yield ascocarps production [17], there are still high fluctuations within the same plantation, resulting in productive and non-productive areas or “patches” [18]. The analysis of ecology, phenology, and interannual fluctuations on mycelial development are also essential for the proper management of mycorrhizal plants producing truffles or mushrooms [19,20,21].

Molecular strategies such as restriction fragment length polymorphism (RLFP), denaturing gradient gel electrophoresis (DGGE) or amplified rDNA restriction analysis (ARDRA) [22] have been used for years to track or monitor edible inoculated fungal species in the field or to identify them in other applications and bioprocesses [23,24,25,26,27,28,29,30]. Among PCR-based methods, the use of species-specific primers in a quantitative real-time PCR (qPCR) approach has been widely applied to trace root and soil mycelia of different mycorrhizal fungi-producing truffles or mushrooms with high socioeconomic impact, such as *Tuber melanosporum* [31,32,33], *Tuber magnatum* [34,35,36], *Tuber aestivum* [37,38], *Lactarius deliciosus* [39,40,41,42], *Tricholoma matsutake* [43], and *Boletus edulis* [42,44,45]. In addition, the use of this technique has been used to quantify the presence of mycelium in black truffle plantations, both in the first years to control the establishment of plantations, and to subsequently evaluate its relationship with the productivity of the crop [20].

Soil properties such as pH (acid or alkaline) and the host plant species lead to the fruiting of different species of desert truffle [1,5]. In recent years, several studies on the genus *Terfezia* have been published to clarify and update the phylogenetic relationships among the new species and those already described within the genus [46,47,48,49,50,51,52,53,54]. These studies showed intraspecific genetic variations in the nrDNA-ITS sequence of *Terfezia* spp., including the identification of some cryptic species [55,56,57], in which only molecular data are required and used for species identification [20,56].

Traditionally *T. claveryi* and *Terfezia crassiverrucosa* Zitouni-Haouar, G. Moreno, Manjón, Fortas & Carlavilla have been collected and marketed together in alkaline soils, because no key differences in distribution, host plant, macroscopy, taste, and flavour characteristics can be found [49]. In fact, they are species very similar morphologically and phylogenetically [49]. Consequently, both species share their habitat in plantations and wild areas and have been called “*turmas*” indistinctly by gatherers. For this reason, and from now on, when the term *turmas* is used in this study, we refer to both marketed *Terfezia* species in Spanish alkaline soils (*T. claveryi* and *T. crassiverrucosa)*. The internal transcribed spacer (ITS) region from ribosomal DNA (ITS1-5.8S-ITS2) has extensively been used as a universal DNA barcode marker for *Fungi* [58]. This region was selected to design specific primers for the detection and quantification of *T. claveryi* and *T. crassiverrucosa* DNA in soil by real-time quantitative PCR (qPCR). Thus, the objectives of this study are as follows: (a) design and check a set of specific primers for the quantification of DNA of these *turmas* in soil by qPCR approach; and (b) apply this strategy to determine how mycelium is distributed and spread in a desert truffle plantation.

## 2. Materials and Methods

### 2.1. Environmental Sampling

A desert truffle plantation with mycorrhized *H. almeriense* × *T. claveryi* plants (located in Torre-Pacheco, Murcia, Spain, 37°46′29.7″ N, 0°57′1.2″ W) was used to investigate the mycelium spreading of the inoculated species in the soil. The plantation was established in 2016, and it was extended again twice more with new mycorrhized plants in 2018 and 2019. This plantation started to be productive one year after the first round of planting in March 2017. The plants were established in rows separated 0.75 m and 1 m between rows. A suitable management of the cultivation and watering according to the recommendations described in [5,13,14,17,59] was followed.

In total, 36 soil samples were collected in February 2020 at an equal distance from the surrounding plants and at a depth of 10–15 cm. They were maintained at 10 °C until they were transported to the laboratory and kept at −20 °C until processing. Before DNA extraction, soil samples were dried at room temperature for 24–48 h. As detailed in Figure 1, 18 samples were from the three rows planted in 2016, 12 from the two rows planted in 2018, and 6 from the one row planted in 2019. There was a separation of 2–2.5 m between the samples within each row, and a distance of 1–1.5 m from one row to the next. Moreover, a soil sample was taken as a negative control from a non-productive area outside the plantation, free of *H. almeriense* mycorrhizal plants. During the fruiting season in spring 2020, 3, 19, and 6 ascocarps of similar weights were collected from 2016, 2018, and 2019 planting areas, respectively.

### 2.2. Soil DNA Extraction

Soil samples were carefully sieved through 500 µm mesh to remove any root fragments, stones, or plant material debris. Then, genomic DNA was extracted in duplicate from 0.25 g of each sample, previously well homogenized, using the DNeasy PowerSoil Kit (Qiagen, Hilden, Germany), according to the manufacturer’s instructions. All DNA was eluted in 100 µL of elution buffer (10 mM Tris) and stored at −20 °C until processing. The concentrations of DNA extractions were measured using a NanoDrop ND-2000 Spectrophotometer (Thermo Fisher Scientific, Waltham, MA, USA), and the quality was examined by 260/280 nm and 260/230 nm optical density ratios.

In the same way, DNA extracted from a mixture of 113.1 mg *T. claveryi* active mycelium (T7 strain), from a pure culture in MMN-O liquid medium [60], and 0.1543 g of negative control soil (twice autoclaved), was used for the generation of the standard curve.

### 2.3. Design of Specific Primers for Turmas

ITS-rDNA (ITS1-5.8S-ITS2) sequences of *T. claveryi*, *T. crassiverrucosa* and other desert truffle species from GenBank and RefSeq databases (Appendix A) were used for primers design by two different web-based software programs: ABI PRISM Primer Express v3.0.1 (Applied Biosystems, Waltham, MA, USA) and ProbeFinder v2.50 (Universal ProbeLibrary, UPL, Assay Design Center) (Roche Molecular Systems, Pleasanton, CA, USA). Multiple sequence alignments were carried out using the MUSCLE algorithm [61] to delimit specific regions for optimal primer selection using MEGA X: Molecular Evolutionary Genetics Analysis across computing platforms v10.0.5 [62].

Customized primer criteria were established according to the following SYBR Green qPCR assay requirements and recommendations [63,64,65]: melting temperature 55–62 °C (opt. 60 °C), GC content 40–55% (opt. 50%), primer size 15–30 nt (opt. 20 nt), amplicon size 75–150 nt (opt. 100 nt), and GC clamp 1 nt. Primer sets generated were examined for cross- and self-dimers and hairpin formations (Beacon Designer software, PREMIER Biosoft’s, Palo Alto, MA, USA). Those with ΔG values of −3.5 kcal/mol and below were avoided. Moreover, amplicon checking for secondary structures was carried out using the UNAFold web tool (IDT, Integrated DNA technologies, Coralville, IA, USA), adjusting the Mg concentration to 3 mM. All structures formed had to meet a Tm (melting temperature) less than the qPCR annealing temperature and values of ΔG above −9 kcal/mol. Furthermore, the oligonucleotides and the obtained amplicons were evaluated in silico for specificity using Megablast search at NCBI GenBank database (https://blast.ncbi.nlm.nih.gov/Blast.cgi; accessed on 1 April 2020) [66].

Direct PCR amplifications from dried ascoma of fungal reference materials (Table 1) were performed in a FlexCycler (Analytik Jena GmbH, Jena, Germany) according to the protocol described by Bonito [67]. Each 25 µL reaction volumes was amplified with ITS1F-ITS4 primer pair [68,69] and it was composed of 0.4 mM for each primer, 0.2 mM for each dNTP, 2.0 mM MgCl_2_, 50 mM KCl, 20 mM Tris-HCl (pH 8.4), 0.04% BSA and 1.25 U of Taq DNA polymerase (Invitrogen). The parameters of the thermal cycler were: initial denaturation for 2 min at 94 °C, 40 cycles consisting of 30 s at 94 °C, 30 s at 55 °C, 1 min at 72 °C, and a final extension for 5 min at 72 °C. PCR products were purified using the EZNA Cycle-Pure-Kit (Omega Bio-Tek), according to the manufacturer’s instructions, and sequenced at the Molecular Biology Service of the University of Murcia. In order to check in vitro specificity, DNA extracts of different species of desert truffles were used as templates (Table 1) under qPCR conditions.

### 2.4. Quantitative Real-Time PCR Conditions

A standard curve was generated from 1/10 dilutions of purified DNA standard (amounts of *T. claveryi* mycelium in soil) with nuclease-free water. Then, the efficiency of the real-time PCR was calculated for each primer pair selected from the value of the slope of the calibration curve [70] (generated as: E = (10^(−1/slope)^ − 1) × 100), and the primer concentration was optimised in the range of 50 to 200 nM for the chosen combination of primers. In addition, the minimum amount of mycelium detected by this qPCR protocol was established.

Real-time SYBR-Green-dye-based PCR amplification was carried out for in vitro tests and experimental samples in 96-well plates using a QuantStudioTM 5 Flex (Applied Biosystems, Waltham, MA, USA) instrument. Each amplification was performed on 10 µL reaction volumes containing 5 µL of Power SYBR Green PCR Master Mix (2×) (Thermo Fisher Scientific), 0.1 µL of each primer at 10 µM, 3.8 µL of nuclease-free water, and 1 µL of 1/5 diluted DNA template. The thermal cycle protocol was 50 °C for 2 min and 95 °C for 10 min at hold stage followed by 40 cycles of 95 °C for 15 s and 60 °C for 60 s at PCR stage. After that, melting curve analysis was used to delete from the analysis those samples with non-target sequences and secondary structures. Three replicates for each standard DNA dilution, for each sample and for a no template control (NTC), were included for each run. Then, C_T_ (cycle threshold) values were automatically converted to quantities of *turmas* mycelium in soil (mg mycelium/g soil) by QuantStudio Design & Analysis software v1.4.

### 2.5. Statistical Analysis

Statistical analyses were performed using the *stats* package in the R software environment (https://www.R-project.org/; accessed on 20 May 2022) [71]. Soil mycelium data were evaluated by Grubbs’ test to determine whether one of the values was a significant outlier from the rest (https://www.graphpad.com/quickcalcs/Grubbs1.cfm; accessed on 20 May 2022). Differences among groups of samples were compared using Kruskal–Wallis tests with the *kruskal.test* function. When the test was significant, post hoc analysis was performed using the *dunnTest* function in the *FSA* package [72]. Correlations between soil-detected mycelium and harvested truffles were analysed by Poisson regression using the *glm* function.

## 3. Results and Discussion

### 3.1. In Silico Primer Screening

The ITS rDNA region is the most commonly used fragment for fungal species identification and as a target for soil fungal diversity studies; however, it shows different intraspecific variability in all groups of fungi and high length polymorphism [58,73,74]. In addition, even though many mycologists advocate LSU region as alternative, the ITS region shows greater efficiency in species discrimination [58]. The consensus sequence was generated from the independent *turmas* sequences (Table 1) aligned by MEGA X software. This sequence was used as a DNA template, resulting in three sets of designed primers (Table 2) based on in silico analyses. The specificity of the primers and the amplicons produced was also confirmed against the sequences of GenBank and RefSeq databases (Appendix A).

ITS regions from multiple alignments of *turmas* and desert truffle sequences showed short and limited sections located within the ITS2 region for the optimal design of specific primers. This made it difficult to obtain primers automatically, and only the primer set TerclaF1/R1 was generated by ProbeFinder software. Moreover, some of the considerations for proper primer composition made the design even more complicated, because when SYBR Green dye is used as fluorescence marker, the presence of primer dimers, the formation of secondary structures, or non-specific amplifications may induce the detection of false signals [64,75]. All this forced the manual design of the primer set TerclaF2/R2 and primer set TerclaF3/R1, using the parameters already set as closely as possible.

### 3.2. Selection and Validation of qPCR-Specific Primers

In vitro specificity was also confirmed for the three set of primers designed, and non-amplifications were found for other fungal species (Table 1). However, the set TerclaF3/R1 provided lower Ct values, with the same amount of *turmas* DNA template as the sets TerclaF1/R1 and TerclaF2/R2. Careful focus was taken with non-specific amplifications of other desert truffle species (*T. albida*, *T. grisea*, *T. eliocrocae*, *Picoa* sp. and *Geopora* sp.), because they can share the habitat and the host plant with *turmas* [1,2,47,56,76]. Moreover, other *Terfezia* species from acid soils under non-*Helianthemum* sp. host plants were tested for cross-validation.

DNA serially diluted of the standard sample (10-fold dilutions) were performed and a calibration curve was constructed from 10^−1^ to 10^−5^ dilutions for three sets of primers designed. The results showed the highest efficiency for primer set TerclaF3/R1, 89% (Figure 2), followed by primer set TerclaF2/R2 and TerclaF1/R1 (64% and 58%, respectively). Moreover, coefficients of determination (R^2^) were always greater than 0.99 in all curves.

Finally, the primers combination chosen were TerclaF3/R1 for optimal real-time qPCR assay using SYBR green fluorescence dye, and they were used for subsequent analyses. In addition, the primer concentration was adjusted to 100 nM, and PCR inhibitors were observed when using pure soil DNA extraction as DNA template. Thus, 1/5 dilutions of each soil DNA extraction were sufficient to avoid inhibition in qPCR reactions. This was an important check point in order to prevent a drop in the efficiency of the samples analysed [70].

The minimal fungal biomass that could be reliably detected was set at 4.23 µg mycelium/g soil, because below this value, reproducibility was lost (Figure 2). Sensitivity levels were different to a greater or lesser degree for other ectomycorrhizal fungi, due to the different strategies used for standard DNA and calibration curve. The detection limit for extraradical mycelium of the edible fungi *L. deliciosus*, *Rhizopogon roseolus* and *Rhizopogon luteolus* was 10-fold lower (0.48 µg mycelium/g soil) from the DNA extraction of fresh mycelium in soil [40]. However, in a previous study, *L. deliciosus* was detected at up to 2 µg mycelium/g soil [39], and *B. edulis* was detected at around 39 µg mycelium/g soil [44]. Later, minimal quantities of *L. deliciosus* and *B. edulis* fungal biomass were detected: 1 and 4 µg mycelium/g soil, respectively [42]. In cases where pure in vitro culture of mycelium is difficult to achieve, such as in *Tuber* species [77], immature ascocarps have been used for standard DNA extraction [32,33,34,38]. Gryndler et al. [37] linked ITS rDNA copies in the PCR product with the biomass of *T. aestivum* mycelium for absolute quantification; however, this method has been questioned for comparison studies because there is a large variability in the number of copies of this gene between fungal species [78].

Real-time qPCR protocols could also be affected by the DNA extraction process, in which the quality of the experiment varies depending on the amount of DNA obtained and contaminants co-extracted [79,80]. However, researchers have commonly added control soil to the extraction DNA procedure in order to generate site-specific calibration curves [32,35]. Furthermore, although TaqMan-based qPCR assays that include hydrolysis probes avoid the detection of non-specific products, SYBR-Green-dye-based techniques have shown the same high-performance results when appropriate qPCR protocols are followed [63,81].

### 3.3. Spatial Dynamic of Turmas Mycelium in a Desert Truffle Orchard

A four-year-old desert truffle orchard was sampled for environmental validation of the primer pair selected, TerclaF3/R1. Mycorrhized plants, inoculated with *T. claveryi* spores, were planted in three different years (2016, 2018, and 2019) (Figure 1); therefore, mycelia from three different ages could be cohabiting. Soil samples were collected in winter, before the fruiting season (spring) of desert truffles in the Mediterranean area [1]. Moreover, winter is the plant’s physiology stage of maximum activity over the year [15]. *H. almeriense* shows a high photosynthetic rate and gas exchange together with a vigorous vegetative growth and flower bud production [15].

Mycelial distribution in the plantation is shown in Figure 3, in which a high variability in fungal biomass in soil between the samples can be appreciated. The range of fungal biomass detected and quantified was from 0.079 to 4.798 mg mycelium/g soil, and only 2 of the 36 samples were undetected. The specificity was also confirmed through checking melting curves after PCR cycles.

No differences in soil fungal biomass were found between years of the planting area (Kruskal–Wallis chi-squared = 0.7417, df = 2, *p*-value = 0.6901) (Table 3). However, significant differences were found between the sampling points (Kruskal–Wallis chi-squared = 12.188, df = 5, *p*-value = 0.0323) (Table 3).

In contrast to the idea of finding a pattern over the years, we detected a heterogeneous mycelial spreading in that plantation, which does not seem to respond to the year of planting. In winter, *turmas* mycelium may concentrate in those areas where the plant requires nutritional support, either due to sub-optimal soil conditions or due to increased plant needs. Moreover, agroclimatic parameters may also have an effect on mycelial development.

In accordance with the desert truffle life cycle [82,83], the first rainfalls of late summer and early autumn promote primordia formation, which is associated with the high production of ascocarps in the next spring [14,15]. As well as in the genus *Tuber* [84,85], *T. claveryi* exhibited a heterothallic lifestyle, which requires the combination of mycelia with different mating type genes in order to form fruiting bodies [82]. Thus, another reason for the heterogeneity of soil mycelium detected in our case study could be related to the mating type mycelia frequency across plantation, but there are still no studies in desert truffles and, thus far, no clear evidence has been found in the genus *Tuber* either. The detection of both mating types was correlated with productive trees in black truffle plantations [86,87], but in other studies, mating type frequency was extended randomly across plantation, and it was not significantly related to black truffle ascocarps harvested [88,89].

The amount of fungal biomass in winter showed no significant relationship with the amount of ascocarps harvested by year (3, 19, and 11 ascocarps collected in 2016, 2019, and 2018, respectively; Poisson regression *p*-value = 0.573). It seems that the middle age section accumulated higher desert fruiting bodies. Although the plantation was very successful in coming into production in the first year after planting, it did not reach its maximum productivity, which would occur around the eighth year [14]. In natural and cultivated experiments of black truffle, significant differences were found between mycelial abundance and productive areas [31,32,33,87]. *T. magnatum* mycelium was significantly higher in the surrounding fruiting areas [35]. Other ectomycorrhizal fungi, such as *L. deliciosus* and *B. edulis*, showed a non-correlation between the productivity of the different plots with the soil fungal biomass [42], but the soil mycelium was strongly related to climatic parameters. The results derived from this assay should be analysed with prudence, because long-term annual studies across season are necessary to explain the mycelial behaviour of desert truffles in soil, as discussed above for other mycorrhizal fungi.

Although the plantation of our study was irrigated and the competing vegetation had been eliminated, other strategies, such as mechanical tilling practices [90,91,92], should be investigated in order to maintain a youthful plantation and avoid the loss of crops after a few years. Therefore, it is necessary to explore the mycelial dynamics over time, because the plantation may be at risk of ageing, with mycelium and mycorrhizae displaced and no ascocarp production.

## 4. Conclusions

In conclusion, the selected primers designed within ITS regions are sufficiently accurate to develop a real-time qPCR protocol for the quantification of fungal biomass of *T. claveryi* and *T. crassiverrucosa* in soil samples. The TerclaF3/R1 primer set was tested and validated for the SYBR-Green-based qPCR assay. Moreover, the preliminary study on soil samples from a desert truffle plantation showed no correlation between winter soil fungal biomass and truffle productivity in spring. However, the amount of fungal soil mycelium seemed to trend to decrease with the years, indicating that, after a certain time, the plantation could become unproductive. In-depth knowledge of mycelial dynamics over the years would help us to develop proposals for plantation management to extend the useful life of plantations.

## Figures and Tables

**Figure 1 jof-08-01095-f001:**
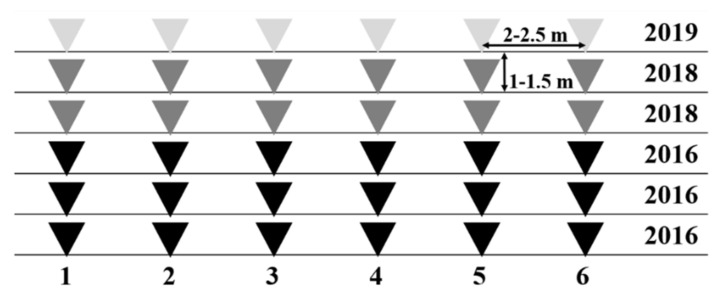
Diagram of sampling points (pyramid marks) in different years of plantation establishment.

**Figure 2 jof-08-01095-f002:**
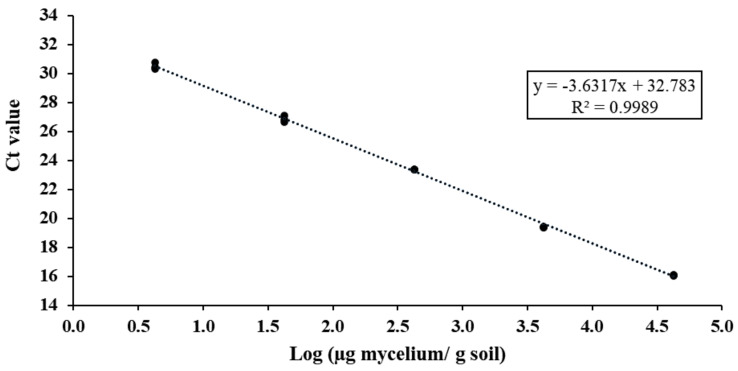
Real-time qPCR standard curve for *T. claveryi* DNA quantification in soil. The curve was generated by plotting the Ct values obtained from 10-fold serial dilutions of DNA standard sample against the logarithm of the quantity of mycelium in soil (μg/g). Efficiency for the primer set TerclaF3/R1 was 89%.

**Figure 3 jof-08-01095-f003:**
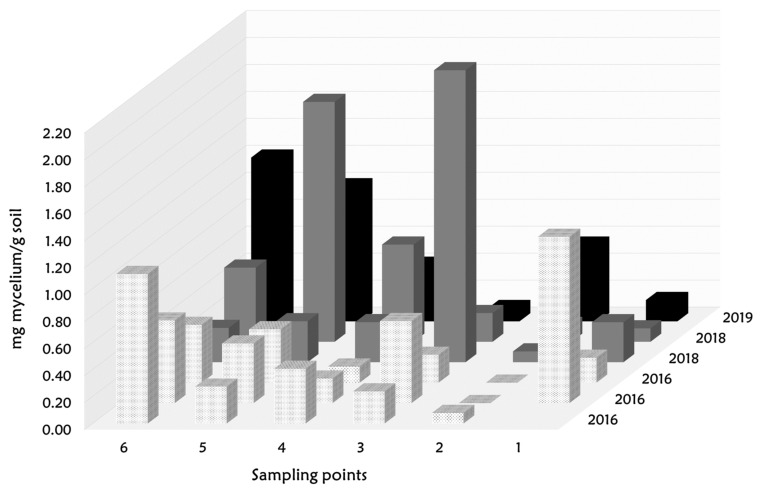
Distribution of *turmas* mycelium in the different planting years (2019 in black, 2018 in grey, and 2016 in white raster) across sampling points (1–6).

**Table 1 jof-08-01095-t001:** Fungal reference materials used in this study.

Taxon	Specimen ID ^1^	GenBank Accession Number
*Terfezia albida* Ant. Rodr., Muñoz-Mohedano & Bordallo	j574	OP458226
*Terfezia eliocrocae* Bordallo, Morte & Honrubia	j579	OP458228
*Terfezia olbiensis* (Tul. & C. Tul.) Sacc.	j588	OP458229
*Terfezia claveryi* Chatin	j592	OP458224
*Terfezia claveryi* Chatin	j596	OP458223
*Terfezia claveryi* Chatin	j597	OP458222
*Terfezia claveryi* Chatin	j216	OP458220
*Terfezia claveryi* Chatin	j73	OP458219
*Terfezia crassiverrucosa* Zitouni-Haouar, G. Moreno, Manjón, Fortas, & Carlavilla	j53	OP458218
*Terfezia crassiverrucosa* Zitouni-Haouar, G. Moreno, Manjón, Fortas, & Carlavilla	j235	OP458221
*Tirmania pinoyi* (Maire) Malençon	j601	MG920185.1
*Tirmania nivea* (Desf.) Trappe	j590	OP458225
*Terfezia grisea* Bordallo, V. Kaounas & Ant. Rodr.	j485	KP189333
*Terfezia fanfani* Mattir.	j484	OP458230
*Terfezia pseudoleptoderma* Bordallo, Ant. Rodr. & Muñoz-Mohedano	j478	OP458231
*Terfezia arenaria* (Moris) Trappe	j466	OP458227
*Terfezia boudieri* Chatin	j371	OP458234
*Tirmania honrubiae* Morte, Bordallo & Ant. Rodr.	j366	OP458233
*Terfezia fanfani* Mattir.	L14	HM056219
*Terfezia extremadurensis* Muñoz-Mohedano, Ant. Rodr. & Bordallo	j96	OP458232
*Terfezia pini* Bordallo, Ant. Rodr. & Muñoz-Mohedano	j151	OP458235
*Picoa* sp. Vittad.	j442	OP458217
*Picoa* sp. Vittad.	j17	OP458215
*Picoa* sp. Vittad.	j59	OP458214
*Picoa* sp. Vittad.	j41	OP458213
*Picoa* sp. Vittad.	j45	OP458212
*Picoa* sp. Vittad.	j20	OP458216
*Geopora* sp. Harkn.	R21b	OP458210
*Geopora* sp. Harkn.	R23	OP458209
*Geopora* sp. Harkn.	j121	OP458211

^1^ Herbarium of University of Murcia (MUB-FUNGI).

**Table 2 jof-08-01095-t002:** Set of primers designed and tested.

Primer Set	Sequence (5′ → 3′)	Length (nt)	Tm (°C)	GC (%)	Amplicon (nt)
TerclaF1 TerclaR1	ATAGGGCATGCCTGTCTGAG	20	60.0	55	106
	TGGAGGGCAACTTAATACACAGT	23	59.2	43	
TerclaF2 TerclaR2	TAACTGTGTATTAAGTTGCCCTCCAG	26	59.0	42	120
	GAGTTGAGGCAAGTACAATCAATCATAC	28	59.2	39	
TerclaF3 TerclaR1	GCTCCCCCTCACTCAAGTAT	20	59.1	55	79
	TGGAGGGCAACTTAATACACAGT	23	59.2	43	

Tm = melting temperature; GC = guanine–cytosine.

**Table 3 jof-08-01095-t003:** Mycelia detected in soil for each variable in desert truffle orchard.

Variable	Samples (N)	Mean Fungal Biomass (Mg Mycelium/g Soil)	SD	Significance Level (*p*-Value < 0.05)
Year 2016	18	0.386	0.350	a
Year 2018	12	0.574	0.684	a
Year 2019	6	0.577	0.451	a
SP-1	5	0.394	0.474	ab
SP-2	6	0.142	0.215	a
SP-3	6	0.588	0.792	ab
SP-4	6	0.358	0.214	ab
SP-5	6	0.700	0.592	b
SP-6	6	0.695	0.383	b

SP: sampling point; SD: standard deviation.

## Data Availability

All sequences used during this experimental study are available from GenBank and accession numbers are given in Table 1 and Appendix A.

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
