# Peer review of "Design and Validation of qPCR-Specific Primers for Quantification of the Marketed Terfezia claveryi and Terfezia crassiverrucosa in Soil"

_jof, 2022, doi:10.3390/jof8101095_

Round 1

Reviewer 1 Report

The manuscript is focused on the development of a DNA based assay for the identification and quantification of a fungus of high economic interest, even if of limited diffusion. The authors, while applying classic and well-known molecular methods, have highlighted how these can be useful to support the complex cultivation of high-value mushrooms. The study is correctly conducted and clearly presented. The materials used are complete to substain the data obtained. In my opinion, this study deserve to be published in JoF because pertinent and of practical interest.

Reviewer 2 Report

# Manuscript: jof-1951950

# Version: peer-review-v1

# ------------------------------------

# Theme

>> The paper is about designing qPCR primers for genus group which includes two species of truffle fungi. The manuscript is of interest and practical use for the one who are interested in subject.

>> There are few minor issues which need to be fixed and a bit more justification needed for using “sensu lato” for two different species under one species name. Sensu lato at higher taxonomic level is fine, but at species level, it is not that common and not without a proper facts and justification.

# ------------------------------------

>L11: either change the word crop or where these since the latter refers to the crop and not the fungus.

>L28: Italicize Pezizaceae

>L29: habitats

>L44: high

>L46-50: too long sentence

>L53: monitor

>L55: species

>L60: what authors and which technique?

>L60-62: needs rewriting.

>L66: On what basis old and new are defined? Was it based on phylogenomics?

>L67: nrDNA-ITS.. what? Add sequence after it.

>L71: New species in Terfezia genus

>L76: How similar these two species are at genome/DNA level? Or how close are they at rRNA gene/ITS level? Can these two different species be called as sensu lato? I wonder.

>L77: change ribosomal DNA to ribosomal RNA gene.

>L87: What kind of co-ordinates are these? What is O? Please change the degree symbol here and with ºC.

>L102: Change “at” with “during the”.

>L119: “ (twice autoclaved)” is used at wrong place.

>L123: The sequence accession are not just from GenBank some are from RefSeq as well. Please fix the issue.

>L127: When you say species specific region, that means the primer must not be for sensu lato.

>L139-140 citation at wrong place.

>L143: wrong style of citation, here and many other places. Please fix.

>L191-193: This information is somewhat different as it was previously described in M&M. Please fix it.

>L194: wrong placement of (table 2)

>L236-237: the sentence is not giving correct meaning, please rewrite.

>L238: what is “ECM fungi”?

>L239-242: This information need clarification about its origin. If not provided, it should be removed.

>L247: same as L77 & L143.

>L290: this is not correct information, or?

>L290-295: I dont understand how a different genus is directly compared and to this genus without any background or relationship? If something is not find in different genus, it can’t be used to deduce any inference to other genera.

>L296: remove and replace word orchard.

>L320: accurate enough.

>L319: mention the name of species which the primer targets rather than just mentioning sense lato, would be easier for the readers.

Table S1: please fix the acccession issue in the table caption. Also, please mention subspecies/strain or other identifier related information in table S1.
